# Virtual Bone Shape Aging

**Francesco Calivá**                    FRANCESCO.CALIVA@UCSF.EDU
**Alejandro Morales Martinez**    ALEJANDRO.MORALESMARTINEZ@UCSF.EDU
**Sharmila Majumdar**                SHARMILA.MAJUMDAR@UCSF.EDU
**Valentina Pedoia**                  VALENTINA.PEDOIA@UCSF.EDU
*Center for Intelligent Imaging, University of California, San Francisco, United States of America*

## Abstract

We use deep learning to age knee bone surfaces four years. We propose to encode an MRI-based bone surface in a spherical coordinate format, and use these spherical maps to predict shape changes in a 48 months time frame, in subjects with and without osteoarthritis. The experiments show that a 2D V-Net can predict bone surface shape with a mean absolute error of about 1 mm. Our code is available [here](#).

**Keywords:** Bone Shape, Osteoarthritis, Spherical Encoding.

## 1. Introduction

Knee osteoarthritis (OA) is a complex joint disease with a global prevalence approaching 5% (Cross et al., 2014). While the OA development involves all tissues of the knee joint, it is considered to be mechanically driven by load-dependent changes in the subchondral bone (Neogi, 2012). As such, bone shape changes are an appealing outcome target for clinical trials, and patient-personalized bone changes trajectory prediction could be of great interest in understanding the impact of specific intervention strategies. In this study, we aim to explore, for the first time, the usage of deep learning to predict bone shape changes in a time frame of 48 months on subjects with and without OA.

## 2. Methods and Experiments

This study uses data from the Osteoarthritis Initiative study, in which 4,796 subjects were scanned at 7 different time points spanning over 8 years. In our previous work, we used 3D Sagittal Double Echo Steady-State (3D-DESS) Magnetic Resonance Imaging (MRI) data to segment the knee bone using a V-Net model (Milletari et al., 2016), we then encoded bone shape features in a 2D spherical map that was used to diagnose and predict future OA (Morales Martinez et al., 2020). For this study, we used the femur bone spherical maps from three time points (baseline, 12 months, and 24 months) as inputs to a model tasked with predicting the longitudinal bone surface changes at the 72 months time point (Fig. 1). We used 4133 subjects split into 2855/598/680 train/val/test sets. BMI, age, and sex were controlled by testing for statistical independence across the splits. The first of our three implemented approaches was a 2D modified V-Net, as a baseline model (Fig. 1). This model was given the concatenation of the spherical bone encoding of the three initial time points as an input, and it produced the bone shape at a subsequent time point (72 months). For the second approach, we corrupted the input spherical maps with white noise

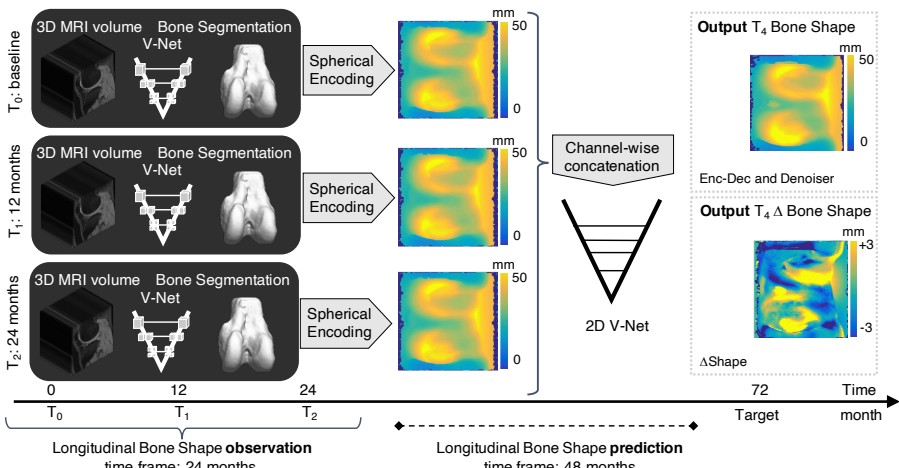

Figure 1: Overview of the proposed approach. All 2D V-Nets consisted of 5 levels with 3, 4, 3, 4, 2 convolutions per-level.

($\mu$=0, var=0.03), and utilized the 2D V-Net as a denoising autoencoder. This solution was chosen as we observed the naive encoder-decoder excessively exploited the input information and merely replicated the latest input time point, which would prevent the network from learning meaningful representations of the data. The final approach was to give the network a more explicit supervision for the bone shape changes. We engineered the network target as the difference between the bone spherical maps at the 24 and 72 months time points. For all the networks, two sets of experiments were conducted. In the first, parameters were tuned by minimizing a mean-squared error (MSE) loss, in the second, a combination of mean-absolute error (MAE) and structural similarity index metric (SSIM) loss, which in our previous work (Calivá et al., 2020) proved instrumental for the restoration of fine details when reconstructing images. In all the models, Adam optimizer with learning rate $1 \times 10-5$, batch-size of 8 samples, 0.05 dropout rate and early-stopping with 15 val-iterations without improvement patience were used.

## 3. Results and Conclusions

We predicted femoral bone remodeling in a cohort of patients with osteoarthritis and control groups 48 months ahead of time. Table 1 shows model comparisons on the whole test set and stratified by OA status. In general, models trained with combined loss outperformed the others. The best results in terms of MAE were obtained by the $\Delta Shape$ model. A visual example comparing actual and predicted bone shape changes is reported in Fig. 2.

To the best of our knowledge, we are the first attempting to solve this challenging but significant task. Bone remodeling is a precursor of osteoarthritis and being able to characterize remodeling before the manifestation of the disease can greatly impact preventative care as well as diagnosis and patient management. In this proof-of-concept work, optimization of the deep learning framework was not the scope; we aim to investigate alternative architectures and training paradigms in an extended version of this work.

Table 1: Bone shape prediction quality, measured in terms of mean-absolute error (MAE) in mm and structural similarity index metric (SSIM).

| | All (N=680) | | OA (N=39) | | Not-OA (N=641) | |
|---|---|---|---|---|---|---|
| Model | MAE | SSIM | MAE | SSIM | MAE | SSIM |
| Enc-Dec | 1.409 (0.653) | 0.922 (0.021) | 1.496 (0.458) | 0.920 (0.019) | 1.404 (0.662) | 0.922 (0.021) |
| Denoiser | 1.653 (0.648) | 0.926 (0.022) | 1.768 (0.517) | 0.924 (0.019) | 1.646 (0.655) | 0.926 (0.022) |
| $\Delta$ Shape | 1.076 (0.630) | 0.912 (0.024) | 1.080 (0.444) | 0.911 (0.021) | 1.075 (0.639) | 0.912 (0.024) |

A) Trained using $\mathcal{L} = SSIM + 6.7 * MAE$

| | All (N=680) | | OA (N=39) | | Not-OA (N=641) | |
|---|---|---|---|---|---|---|
| Model | MAE | SSIM | MAE | SSIM | MAE | SSIM |
| Enc-Dec | 1.283 (0.641) | 0.925 (0.021) | 1.332 (0.452) | 0.922 (0.020) | 1.280 (0.651) | 0.925 (0.021) |
| Denoiser | 1.853 (0.590) | 0.868 (0.018) | 1.934 (0.445) | 0.866 (0.017) | 1.849 (0.598) | 0.868 (0.018) |
| $\Delta$ Shape | 1.272 (0.721) | 0.911 (0.025) | 1.206 (0.459) | 0.910 (0.021) | 1.276 (0.734) | 0.911 (0.025) |

B) Trained using $\mathcal{L} = MSE$

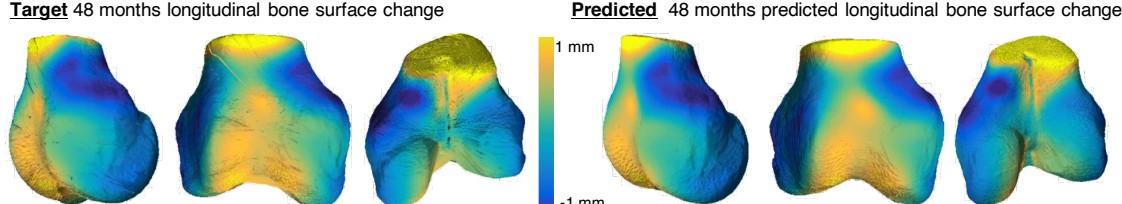

Figure 2: Views of a distal femur, showing the ground truth bone surface change in 48 months (left) and the bone surface change predicted by our model (right).

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
