# OpenReview forum: "Virtual Bone Shape Aging"
_MIDL.io/2021/Conference/Short — MIDL 2021 Poster_

### Official Review · Reviewer_t8UU · 2021-04-21

**Confidence:** 5
**Final Rating:** 3

**Summary:**

The authors are proposing a deep learning method for patient-personalized bone changes trajectory prediction. The method has the potential to predict which patient could develop Knee osteoarthritis. Evaluation is performed on large-scale open-source data. The authors are also promising to make the code available online. The work should be accepted.

**Strengths:**

Large open-source data from the Osteoarthritis Initiative study, in which 4,796 subjects were scanned at 7 different time points spanning over 8 years is used.
The authors will make the code available after acceptance.
The work tried to address an important clinical problem by using off-the-shelf methods. Therefore methodological novelty is limited, however, the clinical application is novel and important.

**Weaknesses:**

The abstract needs more information. Please include some comments about the method, evaluation, and results.
The authors should improve the abstract. Authors should also extend their discussions on different segmentation methods that can be used for segmenting knee bones from volumetric data (3D U net for example)



**Deanonymize Review:**

no

**Detailed Comments:**

See above

**Justification Of The Rating:**

I believe the work is interesting and would generate important discussions during the conference. The authors should improve the abstract. Authors should also extend their discussions on different segmentation methods that can be used for segmenting knee bones from volumetric data (3D U net for example)

**Paper Type:**

validation/application paper

**Special Issue:**

no

---

### Official Review · Reviewer_idaM · 2021-04-29

**Confidence:** 5
**Final Rating:** 4

**Summary:**

In this work, the authors present a method for extrapolating bone shape changes over time. Thus, they propose a generative deep learning framework that utilizes a spherical embedding of the bone surface and predicts the deformed shape based on three observations at distinct time points. Multiple approaches are presented and evaluated to design this prediction and a proof-of-concept study is performed on a large dataset.

**Strengths:**

The authors present an innovative solution for an interesting problem. The spherical embedding idea as well as predicting a delta shape is neat. The results are promising, specifically the reported qualitative ones, and the different approaches are evaluated and compared profoundly. The paper is very clearly and well written and even though a lot of approaches are presented and evaluated, it is easy to follow the red line. Figure 1 helped a lot.
The authors claim to make their code publicly accessible, which is a great plus and follows the open science spirit of MIDL.

**Weaknesses:**

In 3, the authors claim that the delta shape model provided the best results, while this is only the case for the MAE. Regarding SSIM, the other approaches outperform the delta shape method. This should be addressed adequately.

**Deanonymize Review:**

yes

**Detailed Comments:**

As mentioned above, the paper is very well-written and clearly understandable.

**Justification Of The Rating:**

This study is highly innovative, well-conducted, well-written and features a sound evaluation. The presented method is not only relevant for bone aging but could be translated to similar problems as well. Hence, I am sure that this work will lead to insightful and valuable discussions and will complement the program of MIDL 2021. I vote for accepting the manuscript.

**Paper Type:**

both

**Special Issue:**

yes

---

### Meta-Review · Area_Chair_nykq · 2021-05-09

**Recommendation:** Accept (Poster)
**Confidence:** 5

**Metareview:**

Both reviewers praise the quality of this work and recommend acceptance. I agree with them and would only encourage to enhance the abstract if possible.

---

### Decision · Program_Chairs · 2021-05-11

Accept (Poster)